# Life and Quality of Life of Older Adults in Residential Centers in Spain: A Focus Group Analysis Based on Center Quality from the Perspective of Management Stakeholders

**DOI:** 10.3390/healthcare12232446

**Published:** 2024-12-04

**Authors:** Yolanda María De-la-Fuente-Robles, Antonia Rodríguez-Martínez, María del Carmen Martín-Cano, Juan José Jiménez-Delgado

**Affiliations:** 1Department of Psychology, University of Jaén, Campus las Lagunillas s/n, 23071 Jaén, Spain; ymfuente@ujaen.es (Y.M.D.-l.-F.-R.); mmcano@ujaen.es (M.d.C.M.-C.); 2Department of Computer Science, University of Jaén, Campus las Lagunillas s/n, 23071 Jaén, Spain; juanjo@ujaen.es

**Keywords:** quality of life, residential centers, focus groups, older adults, cultural change, person-centered care

## Abstract

Background/Objectives: New residential models for older adults highlight the need for both public and private institutions to adopt new governance and management approaches. These approaches focus on strategic frameworks that guide decision-making, daily operations, and processes aimed at addressing and resolving the economic and social challenges faced by our society while promoting quality standards that serve the public good. The primary objective of this research is to identify and analyze the quality of residential centers and their impact on improving the quality of life of older adult residents from the perspective of management stakeholders; Methods: This study employed a qualitative methodology, involving systematic literature reviews and the analysis of focus groups. The participants were professionals from various fields working with older adults in Spanish residential centers; Results: The results reveal widespread dissatisfaction among participants regarding the need for change to improve both the quality of the centers and the quality of life for older adult residents; Conclusions: This study suggests that the well-being of older adults residing in these centers requires regulatory changes that focus on person-centered care and are tailored to the specific needs of the residents.

## 1. Introduction

The initial premise of this research is based on the natural aging process, a topic that has generated multiple definitions over time due to cultural factors and demographic changes [1,2,3]. According to the National Institute on Aging (NIH) [4], aging is understood as a universal, irreversible, and continuous process that affects all individuals, although its impact varies depending on the social context [5]. In Spain, aging is a growing phenomenon: in 2020, the population aged 65 and over reached 22.9% [6], with an aging index of 125.7% [7,8].

On a global scale, life expectancy has experienced a notable increase in recent decades, primarily driven by improved access to healthcare services, medical advancements, and better living and social conditions. According to data from the World Health Organization (WHO) [9], the global average life expectancy reached 73 years, although significant regional disparities exist. In Spain, the increase in life expectancy is particularly striking, rising from 38.8 years in 1910 to an average of 83.2 years in 2022, as reported by El Economista [10]. Other studies predict that Spain will have the highest life expectancy in the world by 2040, with a significant increase in the number of older adults [11]. Additionally, demographic projections made by the National Statistics Institute [12] highlight the pressure that aging will place on the country, with the population of individuals aged 65 and older expected to represent 26% of the total population by 2037.

As life expectancy continues to rise, there has been an increasing demand for housing and care options for older adults, particularly in residential care centers. These centers are understood as environments where individuals reside for extended periods and receive support with daily living, social, and healthcare needs in supervised and communal settings. The services, approaches, and amenities within these centers vary depending on the population they serve and their levels of independence and autonomy. Internationally, these centers are comparable to care homes.

Residential care centers accommodate individuals with diverse characteristics, such as varying ages, health statuses, autonomy levels, psychological and emotional conditions, and family and social situations, as well as differing abilities, hobbies, and interests. Additionally, they may include residents funded through public or private means. The quality of these centers is influenced by several factors, including staff training and qualifications, empathy and human interaction with residents, organizational and management aspects, care protocols, facilities, hygiene, safety standards, and adherence to regulatory frameworks. These centers must offer comprehensive care that addresses the biological, psychological, and social needs of the residents [13,14]. Studies conducted in other countries also highlight the relationship between management practices and the quality of care [15,16].

The growing demand for residential care spaces for older adults is not only a local issue in Spain but a global challenge. Internationally, some countries have implemented programs aimed at improving the quality of residential care centers [17]. In Spain, research has focused on the well-being of older adults in care homes [18], emphasizing the importance of quality of life (QoL) in these settings [8,19]. The growing older adult population in the country requires an increase in residential care spots, which in 2022 were 4.8 per 100 people aged 65 and over [12].

In response to these challenges, the Ministry of Social Rights and Agenda 2030 approved a new model of care for dependent individuals in 2022, emphasizing personalized care [20]. This regulatory shift, along with continuous demographic changes, makes it imperative to conduct an in-depth analysis of the quality of residential care centers and their impact on the QoL of older adult residents. The evolving demographic and regulatory landscape in Spain is pushing for stronger policy reforms and innovative management strategies within these institutions.

From a regulatory perspective, Spain’s Law 39/2006 [21] establishes the right of dependent individuals to receive care, including specific attention to their assessed needs. However, its implementation has faced numerous challenges, which highlights the importance of addressing both the operational and policy frameworks in the care system [22]. In Andalusia, strategic plans have been devised to improve care for older adults, but their application has been delayed due to the COVID-19 pandemic. Additionally, the regulation of residential care centers in the region is specified in the Order of November 5 [23], which addresses the accreditation and functioning of these centers.

QoL is a central aspect in the care of older adults, particularly within residential care centers. While previous studies have identified various factors that influence QoL, such as autonomy, social participation, and satisfaction with the environment, it remains essential to explore how these factors interact within the context of residential care centers. These centers, which vary greatly in terms of facilities, management, and the level of care provided, directly impact the QoL of residents. Given the increasing aging population and the growing demand for such services, understanding how these facilities contribute to residents’ overall well-being is crucial.

The main objective of this study is to identify and analyze the quality of residential care centers for older adults in Spain, focusing on how it influences the improvement of residents’ QoL from the perspective of management stakeholders. Specifically, the study aims to explore professionals’ perceptions of the factors that impact both the quality of the centers and the residents’ QoL. Through the use of focus groups, the research gathers insights from professionals working in residential care centers, including social workers, directors, and staff, to understand their opinions, beliefs, and perceptions. Additionally, the study seeks to identify areas for improvement in the management and policies of residential care centers, with the goal of enhancing person-centered care and addressing the social and economic challenges posed by an aging population. Lastly, the research analyzes the relationship between the characteristics of residential care centers and the QoL of their residents.

In the context of these objectives, the literature review of Rodríguez-Martínez et al. [24] presents a clear picture of the shortage of residential care spaces in Spain, alongside a detailed analysis of population growth trends in the coming years. It highlights the urgent need for increased residential capacity to accommodate both dependent and independent older adults. However, the current reality reflects a significant shortage of spaces, and future projections suggest this gap will widen, particularly due to the effects of the COVID-19 pandemic on care facilities. This underlines the pressing need for changes in both management and care models to meet the demands of an aging population.

Furthermore, Rodríguez-Martínez et al. [25] identify the multifactorial nature of QoL for older adults in residential centers, emphasizing the limited evidence connecting facility characteristics to residents’ QoL. This review calls for further exploration of these factors to enhance resident satisfaction and well-being. Factors such as the physical environment, equipment management, and organizational resources have been shown to significantly impact residents’ QoL. The studies cited within this review indicate that accessibility, lighting, ventilation, and recreational facilities are positively correlated with QoL, while staffing levels, facility size, and leadership stability are key components influencing both social and psychological well-being.

This body of research underscores the importance of understanding how facility features, staffing, and organizational structure affect QoL, aligning well with the objectives of this study. By investigating the perceptions of professionals working in these centers, the study aims to offer valuable insights into how these factors can be optimized to improve the overall quality of care and life for older adults in residential settings.

## 2. Materials and Methods

This research was conducted using a qualitative methodology, grounded in literature reviews and the analysis of focus groups (FGs) through a thematic analysis approach, guided by a deductive framework [26,27,28,29]. The primary objective was to highlight the data and results obtained from the FGs regarding the quality of residential care centers and the QoL of their residents.

The first step in the research process involved gathering information to provide a comprehensive overview of the current status of older adults and the availability of residential care centers in Spain. This was achieved through the consultation and analysis of the literature review conducted by Rodríguez-Martínez et al. [24], which allowed us to identify future research challenges given the increasing older adult population. Additionally, as the research includes the year 2020, the literature review also considered the circumstances experienced by older adults living in residential care centers during the SARS-CoV-2 pandemic and the emerging cultural shifts aimed at improving the QoL of this demographic.

The aforementioned literature review informed the focus of our investigation, enabling the identification and analysis of studies related to QoL measures and the categorization of residential care center characteristics. The relationship between these characteristics and various QoL measures or their dimensions was systematically analyzed, based on the literature review conducted by Rodríguez-Martínez et al. [25].

After completing the literature review and identifying the key parameters and characteristics to study in our local context, we opted to design and establish FGs using key informants [26]. This research and data collection technique focuses on gathering opinions, beliefs, perceptions, interests, and attitudes from individuals directly involved in our area of study [30].

### 2.1. Participants

In the course of this research, three FGs were formed, targeting the main stakeholders directly involved in the management of residential care centers at the local, regional, and national levels. A sample of 20 participants (*n* = 20) was selected through purposive sampling, representing different management sectors, political profiles, scientific entities, and technical personnel who work daily in residential care centers for older adults.

Recruitment began with the assistance of a key informant who facilitated connections with other professionals, using a snowball sampling method to identify suitable candidates. Inclusion criteria required participants to have at least two years of experience in their respective roles (management, political, or technical), be actively involved in decision-making or daily operations, and possess knowledge of quality-of-life issues affecting older adults in residential settings. Potential participants were identified through professional networks and subsequently contacted via email and phone. Detailed information about the study objectives, procedures, and their rights as participants was provided during the recruitment process. Participation was entirely voluntary, and informed consent was obtained from all participants prior to their inclusion in the study. The response rate was approximately 60%, with 20 individuals agreeing to participate out of 33 contacted. Below, the focus groups conducted as part of this study are described:Focus Group—Management and Technical Staff (FG-MTS): Composed of staff from residential care centers spanning both urban and rural centers, as well as large-, medium-, and small-sized facilities. This FG consisted of seven participants from various residential centers, with a representative sample drawn from different companies or entities, as well as diverse geographic locations.Focus Group—Scientific and Technical staff from Organizations (FG-STO): Made up of staff from organizations working for and on behalf of older adults, managing residential care centers. This FG included eight participants.Focus Group—Politically Profiled (FG-PP): A group consisting of five political officials who currently hold, or have previously held, positions directly related to the care of older adults and residential care management. The configuration of this FG was particularly challenging, as political figures or public officeholders often refrain from participating in research or disclosing potentially sensitive information, particularly regarding older adults and, to some extent, due to the aftermath of the pandemic.

The participants were distributed across three sessions held between 19 and 29 December 2023. All participants were of Spanish nationality, and their sociodemographic characteristics are detailed in Table 1.

### 2.2. Instruments

The primary data collection tool for the FGs was the semi-structured open interview [31]. The working guide for the FGs (Appendix A) was developed based on terms identified in previous studies [24,25]. The main characteristics studied are presented in Table 2.

Additionally, the development of the guide considered the General Inspection Plan for Social Services for the years 2020 and 2021 by the Junta de Andalucía, which consists of five lines of action and 17 programs. Specifically, it took into account the actions related to centers for older adults as outlined in the Annual Execution Report of the Social Services Inspection [32].

**Table 2 healthcare-12-02446-t002:** Characteristics derived from the literature associated with a question from FGs (Appendix A).

Characteristic	Question id	Contribution to Quality of Life (QoL)
Quality certifications	Quality measures	No studies found
Supporting facilities category	Specific support	Positive relationship in some dimensions [33,34]
Large size/number of beds	Size	Significant negative contribution [35,36,37]
Chain affiliation	Affiliation with a chain	Contributes negatively [37]
Space management category	Space management	Positively contributes in some dimensions [33,34]
Building services	Construction	Positive influence from ventilation, lighting, and water supply [34]
Location	Location	Better care in rural areas [37,38]
Occupancy rate	Occupancy ratio	Positively contributes only in some dimensions [36]
Turnover	Staff turnover	Negative influence [39]
Ownership	Ownership	Significantly predicts overall QoL [38]
Private rooms	Typology of places	Significantly and positively contributes [36]
Staff-related category	Dedication of hours	Associated positively with management hours, care hours, and activity hours [17,35,36,38]

Source: Own elaboration.

The information collected from the FGs was recorded in both video and audio formats to ensure accurate data capture and facilitate rigorous content analysis and observation.

An Informed Consent Declaration was designed and provided to participants via email, along with the confirmation of the date and time for the session and FG. This email served as a means of confirming their attendance. The document outlined the rationale for the research and invited participants to accept their informed and voluntary participation. It also emphasized the confidentiality of the data and their exclusive use for research purposes, in accordance with the Declaration of Helsinki.

Transcriptions of FG sessions were completed within 20 days following each FG session. The private web application YouTube was utilized for its ease of use, speed, and reliability in generating transcriptions. Each transcription was subsequently reviewed alongside the audio and video recordings, with personal references removed and participant contributions coded [40].

ATLAS.ti version 23 software was employed for the data analysis and structuring of codes to identify patterns within the data [41,42]. A deductive approach was employed in the analysis, moving from general to specific. We began with the overarching theme of quality in residential care centers for older adults in Spain, subsequently narrowing down the theory by formulating specific questions and gathering information. Finally, we addressed these questions with data and verified or refined the initial premise.

### 2.3. Procedure

The FGs were conducted from 19 to 29 December 2023, via the online platform Google Meet. This format allowed for video and audio recording, ensuring the participation of a geographically representative sample from different parts of the country. The sessions were audio-recorded to capture accurate contributions and avoid the omission of participants’ input. The structure of each FG followed a defined sequence:Introduction and Presentation: The session began with an introduction covering the session procedures and ground rules for each FG. Participants were encouraged to share their views, ideas, and information on various aspects of elder care policies, work methodologies, improvement options, and intervention lines for older adults in residential centers. They were informed of the purpose of this research, specifically to understand the quality of residential centers and, by extension, the QoL of residents. Participants were also advised that, to ensure accurate transcription of contributions, only one person should speak at a time. Each participant was given the opportunity to respond without needing to address every question, allowing them to contribute meaningfully without redundancy.Issue Presentation: In this section, the moderator sequentially presented each question, granting participants the opportunity to speak in the order of their requests, thereby avoiding overlapping interventions and ensuring a smooth session flow.Conclusion, Thanks, and Farewell: The final part included concluding remarks, expressions of gratitude, and a formal closing.

Each FG session was conducted respectfully and with regard to participants’ interests, lasting no longer than 1 h and 45 min. Following the sessions, detailed transcriptions were carried out, anonymizing participants through coded identifiers. This approach allowed responses to be attributed to individual participants while preserving confidentiality and adhering to ethical standards governing scientific research. All data management complied with Spain’s Organic Law 3/2018, dated 5 December, on the Protection of Personal Data and Digital Rights Assurance [43].

A set of questions was developed to capture expert and professional perspectives across different domains within the FGs on key variables in residential centers. These questions also sought to understand participants’ perceptions of how these variables influence center quality and the QoL of older adult residents (Appendix A).

The theme of ownership (Q1) explores how the type of ownership (private vs. public) affects the quality of care and life in residential centers. The typology of places (Q2) theme considers the impact of facility type, such as assisted living or private residential homes, on residents’ well-being. Affiliation with a chain (Q3) examines how being part of a larger network influences care quality and flexibility, while size (Q4) looks at how the physical size of a center affects personalized care and resource availability.

The location (Q5) theme addresses how urban or rural settings impact residents’ accessibility, social integration, and overall well-being. Space management (Q6) focuses on how the layout and functionality of the center’s spaces contribute to residents’ comfort, safety, and independence. Specific support (Q7) explores the availability of cognitive and physical support, such as memory exercises, non-slip flooring, and accessible facilities, enhancing daily life for residents.

The construction (Q8) theme examines the quality of the center’s physical infrastructure, such as ventilation, lighting, and water supply, and how these factors contribute to residents’ comfort and safety. Occupancy ratio (Q9) looks at how the number of residents in relation to available staff affects the quality of care, with higher ratios potentially leading to less personalized attention. Dedication of hours (Q10) highlights the time allocated by staff to administrative, care, and recreational tasks, underscoring the importance of balancing these duties for quality service.

The theme of staff turnover (Q11) emphasizes the impact of staff continuity on care quality, with high turnover potentially disrupting relationships with residents. Finally, the quality measures (Q12) theme addresses the standards and certifications used to evaluate residential centers, influencing care levels and residents’ satisfaction.

### 2.4. Data Processing and Analysis

The audio and video recordings of the FG discussions were transcribed verbatim, ensuring an accurate and complete representation of participants’ contributions. To maintain confidentiality, all transcripts were anonymized, and a random subset of transcriptions was reviewed to verify their accuracy. The qualitative data were systematically organized and analyzed using ATLAS.ti 23 in order to facilitate the coding and thematic analysis process.

Focus group (FG) analysis was conducted based on transcripts from interviews following a structured guide rooted in a prior literature review. Coding, as shown in Figure 1, was applied to various issues identified in the literature review. This coding was then associated with quotes using ATLAS.ti.

A thematic analysis approach was employed, guided by a deductive framework derived from the existing literature on QoL in residential care centers. This framework informed the formulation of specific questions and provided a structured basis for identifying relevant themes and patterns within the data. Initially, the researchers familiarized themselves with the transcripts, generating codes that reflected both predefined categories from the literature and new patterns emerging from the data.

The inclusion and exclusion criteria for data coding were determined by the relevance of text fragments to the research questions and the frequency with which specific themes appeared in participants’ responses. Subsequently, the identified themes were compared with findings from previous studies, enabling a comprehensive analysis that contextualized the results within the broader academic discourse.

To enhance the credibility and validity of the findings, participant feedback was sought during the analysis process. A summary of the preliminary results was shared with participants to obtain their perspectives and validate the interpretation of their responses. Themes were further refined based on this feedback, ensuring that the analysis accurately reflected the views and experiences of the participants.

Finally, representative excerpts from participants were selected to illustrate each theme, providing direct insights into the data and maintaining alignment with the study’s primary focus.

## 3. Results

Based on the objectives and methodology established, the results correspond to each methodological phase developed, detailed as follows.

The analysis draws on the co-occurrence table of question-related codes across the contributions of the three FG groups (Figure 2 and Figure 3). Figure 3 presents a matrix displaying the frequency with which each group (FG-MTS, FG-STO, FG-PP) addressed the questions posed in the study (Q1 to Q12), as well as the total number of interventions per group and question. This information allows us to identify the questions that generated the most interest and debate within each group.

The Sankey diagram (Figure 2) complements the information in Figure 3 by visualizing the flow of interventions from each group in relation to the questions. The thickness of the lines reflects the frequency of interventions, facilitating the identification of the questions that received the most attention from participants. The color of each line corresponds to the color associated with each code in Figure 1.

We can observe, from Figure 3, that the FG-MTS group, consisting of technical and management staff from the centers, devoted more attention to questions related to the dedication of hours (Q10), the size of the centers (Q4), and space management (Q6), reflecting their concerns about working conditions and the organization of the centers. We can also observe that all three groups agree on the importance of dedication and care (Q10), while opinions on the ownership of the centers (Q1) are more varied. The last can be observed from Figure 2 as reflected in the thickness of the lines.

### 3.1. Q1. Ownership and Q2. Typology of Places

A significant issue highlighted in this research pertains to the ownership of the facility, whether public, private, or publicly subsidized, and the type of facility based on the types of available places. This includes considerations of whether residential centers also offer daycare services or slots for individuals with disabilities, noting that some facilities provide slots for both older adult individuals and those with disabilities. Participants from all three FGs indicated that there is no difference in terms of ownership or facility type, as the regulations are the same for all, governed by the number of slots relative to the size of the center.

FG-STO-P2: “[…] the debate in this field always tends to focus on public versus private. I believe we should talk about quality, and quality does not recognize public or private distinctions. In fact, there is no real difference between a public center and a private one; the personnel resources are the same.”

FG-MTS-P7: “[…] I believe that the quality of care for older adult individuals is essentially the same in public and private facilities. In fact, I would argue that private facilities need to be more diligent to avoid negative reputations.”

Responses from the political personnel group reveal significant insights compared to the other groups, particularly regarding the perception that publicly owned centers have larger budgets than privately owned residential care centers. This difference is considered crucial for enhancing the care and QoL of residents. However, they indicate that the quality of services in public centers is significantly poorer:

FG-PP-P1: “[…] publicly owned centers have significantly lower quality than privately owned ones, significantly […]”

FG-PP-P1: “[…] publicly owned centers have a much larger budget; the cost of assistance per euro per person is much higher in public centers than in private ones […]”

In general, no significant differences are perceived in the quality of care or the quality of life of residents based on the typology of places. This uniformity in care quality is attributed to staff management. The same professionals provide care to residents with placements of varying characteristics, without distinction in dedication or level of care.

FG-MTS-P1: “The places are arranged with the Junta de Andalucía. I have 30 for assisted living, 5 for severe behavioral disorders, and 2 private placements. In addition, I have 20 daycare places. I have quite a mix here, and the quality [...] well, I’m the first to set high standards. If I’m demanding, then others can be demanding with me too. […] my residents are very demanding [...]”

FG-MTS-P5: “In my center, there are no assisted living residents, but they would be treated the same based on their needs, especially considering what they pay in a center like this. In other centers, we had all types of residents, and the level of care doesn’t decrease whether they are assisted or independent. The difference lies in that the care of assisted residents requires more time, and that falls on the staff.”

### 3.2. Q3. Affiliation with a Chain

In addressing another set of issues, we inquired about the perspectives each group holds regarding the management of centers operated by entities belonging to the same network or business chain. The general consensus was favorable, indicating that being part of an organized business chain provides security, support, and backing for the workers:

FG-MTS-P4: “[…] as employees, we have the backing of a large company, […] workers are better protected by our organization.”

FG-PP-P1: “[…] the level of professionalism exhibited by some groups compared to others is evident […]”

### 3.3. Q4. Size

Across all groups, the size of residential centers was highlighted as one of the most critical factors affecting resident care. It is not merely the availability of publicly funded or contracted placements, or the presence of facilities for highly dependent individuals or day centers, but also the care provided in larger facilities where residents may not know their fellow inhabitants, sometimes even those they share a room with, which compromises their privacy. Participants from various groups unanimously expressed a preference for smaller centers or for the unification of living units. This sentiment has been underscored by the COVID-19 pandemic, which has presented numerous challenges while also offering valuable lessons. For instance, smaller living units facilitate better control over easily transmissible diseases. Furthermore, the preference for smaller units extends to considerations for residents, families, and staff alike, emphasizing the importance of living arrangements based on compatibility, physical or mental abilities, and cognitive needs. Such arrangements enable the design of personalized life plans, fostering a meaningful and fulfilling existence for each individual.

FG-MTS-P4: “[...] the issue of care primarily lies with the workers, particularly the aides who are constantly moving between tasks. In most cases, they may not even know the names of the residents, especially in a facility with 192 beds, where new residents are frequently admitted. Furthermore, our center is almost always at full capacity, which means there are many residents and staff […]”

FG-MTS-P6: “In my center, which accommodates 36 older adults, the environment is much more familial. We can sit with them for a while, take a walk if they are anxious. It’s more focused on the individual, whereas in a larger facility, it’s challenging to meet those needs […]”

FG-STO-P2: “[...] the size truly influences the care provided; I believe we should move away from those enormous macrocenters towards more controlled environments […]”

### 3.4. Q5. Location

The overall response from the analyzed sample indicates that the location of a center is crucial for the development of the residents’ life projects. This is particularly pertinent considering that individuals wish to continue living in familiar environments and interacting with people they have known for years. They want to stroll in their usual parks and continue greeting their neighbors and friends. Unfortunately, these considerations have not been incorporated into the urban planning of localities. Consequently, residential centers for older adults are often located on the outskirts of urban areas, and even when situated in central locations, they frequently face accessibility issues (e.g., parking, ambulance access, etc.).

FG-STO-P5: “I do consider location to be very important because there was a trend to move residential care centers to the outskirts of cities. I believe the residence should remain in its community. Regarding single rooms, privacy has been well demonstrated. I also advocate for another model [...]”

FG-PP-P5: “I believe that care facilities should be integrated into society, but for that to happen, it’s not just about the location of the facility. It’s also necessary to consider what programs or measures are implemented to bring the community into the facility and to integrate the facility into the community [...]”

### 3.5. Q6. Space Management

Overall, all participants emphasized that their experience during the COVID-19 pandemic highlighted the need for more versatile spaces, moving away from the rigid constraints of current regulations. However, compliance with regulations remains important, particularly in ensuring the proper identification and signage of spaces to meet the needs of residents.

FG-MTS-P3: “[...] we had to adapt like everyone else, but now, practically everything is back to normal, and each space is used for its intended purpose. Spaces are well identified, and signage is in place so that residents and their families can easily recognize them. Then, when the inspection comes [...] it’s all sorted.”

FG-STO-P4: “[...] how these spaces are designed and how they are used is significant. The environment, whether it feels welcoming or like a home model, impacts many aspects, including the encouragement of meaningful activities for residents. Because if I am in a sterile place, where nothing reminds me of my life, it doesn’t help me or invite me to engage in participatory activities related to my life. But if I am in a space that feels familiar, like my home, with objects that suggest meaningful activities for me [...]”

### 3.6. Q7. Specific Support

A significant area of contention and lack of consensus arose concerning the use of specific supports for residents, and how this may influence the QoL of these individuals. A particularly important and controversial topic was the use of physical and chemical restraints; therefore, the focus groups concentrated on this issue instead of considering other specific supports for residents, such as cognitive support or accessibility. The issue lies in the fact that regulations both mandate and restrict, positioning restraints and, for example, medication, in the same framework. In this regard, participants expressed that this topic should be explored more thoroughly by the authorities, particularly concerning medication, medical treatment, and chemical restraints:

FG-MTS-P1: “Not having any type of restraint, while misunderstanding the term restraint, I know there are individuals, particularly those with intellectual disabilities, who require medications. Ultimately, it’s unclear to what extent this medication constitutes a treatment and to what extent it represents a form of restraint; I believe this is an issue that needs to be studied in greater depth, with input from professionals.”

FG-MTS-P2: “In the past year, I agree that, especially regarding what we can refer to as restraints, a generalized approach cannot be applied as it might have been 15 years ago. However, in the last year, we have transitioned to a point where practically all centers in Andalusia could be labeled as restraint-free due to the directive issued by the prosecutor’s office and the instructions we received from the administration in the past year.”

### 3.7. Q8. Construction

A recurring theme in the focus group discussions was the impact of the infrastructure of residential care centers on the QoL of older adults. Participants frequently highlighted the challenges posed by outdated buildings, emphasizing how the age and design of these structures can influence the overall experience of residents. These observations underline the need to update infrastructure in residential care centers to ensure they meet modern standards, both in terms of compliance with regulations and in enhancing the QoL of residents.

FG-MTS-P6: “[...] the facility I’m currently in is very old, and while some things are in good condition because we have to comply with regulations and all that, there are other aspects that, of course, are affected by it being an old building, and that makes a big difference.”

FG-PP-P3: “[...] additionally, many of these facilities are often old, with outdated buildings, which ties into something you mentioned earlier and also has a negative impact.”

### 3.8. Q9. Occupancy Ratio

The occupancy ratio sparked controversy among participants in the various focus groups, as it remains a highly debated topic in our country today and is one of the primary demands in new residential care models. Regarding this issue, the discussion on occupancy ratios in residential centers is dictated by legal stipulations. Ultimately, residential centers in our country operate at 100% capacity.

FG-MTS-P2: “[...] the occupancy ratio is determined by the regulations for each center; that’s what we adhere to.”

FG-MTS-P4: “[...] it cannot be the same for my center, which is a high-capacity facility and always at full occupancy, as for smaller centers, which, for example, might have less staff turnover or residents with shorter stays. In the end, there are many factors that the law does not take into account.”

### 3.9. Q10. Dedication of Hours

The most significant findings pertain to dedication and care, specifically regarding the number of hours dedicated to resident care and support compared to administrative tasks. Across the three groups, there was consensus that in larger facilities with a higher number of residents, the quality of care and individual attention to older adult residents tends to decline. This often results in diminished opportunities for meaningful listening and understanding, as well as a lack of support for planning personal life projects, as illustrated by the following response:

FG-PP-P5: “[…] To begin with, in a 50-bed facility, there is one social worker who knows the entire residence; in a 190-bed facility, there are three social workers, each knowing a specific group of residents. Ultimately, this dilutes one’s ability to truly understand and delve into each resident’s social needs.”

The general trend across all groups emphasizes that the quality of care remains consistent across both public and private facilities, given that the same personnel manage both. There is no separation of staff dedicated to residents occupying private or public spaces within the same facility; hence, the attention, dedication, and level of care provided remain uniform regardless of ownership:

FG-STO-P2: “[…] the personnel resources and quality of care are the same in both public and private facilities, shared across residents; what matters is the dedication and attention that staff offer to residents […]”

However, there is one group, the management and technical staff, that expressed concern about the significant amount of time they devote daily to bureaucratic and administrative tasks. They face numerous requirements and processes imposed by regulations and the necessity for effective administrative management mandated by authorities, particularly during inspections. This situation necessitates extensive desk work, which limits the time available for interacting with residents, especially in larger facilities with many residential slots. Additionally, they mentioned that the hours spent on administrative tasks increase when they have students or interns from various professions and disciplines, further restricting their time.

FG-MTS-P3: “[…] administrative tasks are clearly necessary; we need to document everything and ensure everything is in order because inspections demand it. When we do that work well, it makes the process smooth when inspectors come. However, I understand that much of what we do is unnecessary.”

FG-MTS-P6: “[…] I think we live on the edge, in terms of care. It’s not that residents aren’t attended to; rather, we always seem overwhelmed regarding care. You can’t sit with a resident for five minutes to comfort them or see why they are anxious. Thus, I believe that care tends to suffer more in larger facilities with many residents compared to smaller ones, where interactions are more personal and familial.”

### 3.10. Q11. Staff Turnover

Regarding the issue of staff turnover and continuity with residents over time in residential centers, participants from all three FGs almost unanimously expressed that COVID-19 has taught us valuable lessons, particularly in controlling and preventing the spread of certain diseases. They emphasized that maintaining regular contact is crucial for providing better care, especially concerning specific illnesses and physical decline, as well as for developing a deeper understanding of residents’ life histories.

FG-STO-P7: “The topic of staff rotation is interesting; we learned not to rotate staff due to COVID to prevent contagion, when the real importance lies in having a familiar worker for each resident. It’s ironic that it took a virus to remind us to do this, as it’s essential for the older adults to always have their reference caregivers. However, we learned this lesson through much suffering.”

FG-STO-P6: “[...] staff members need to know the residents and understand their histories to tailor their care appropriately. Their focus should be on the residents’ needs rather than the administrative issues at hand.”

### 3.11. Q12. Quality Measures

We finally raised the question of what measures are used to evaluate the quality of a center. We mentioned quality certifications and the development of Corporate Social Responsibility (CSR) practices as examples. Participants particularly stressed that quality standards should be established by public administration, taking into account the opinions of residents through participatory bodies within the residential centers. This approach would help determine the centers’ quality standards and ensure transparency in disseminating this information, enabling residents and their families to make informed decisions about their admission to a center.

FG-MTS-P2: “It is crucial to guarantee citizens access to quality care and to discuss perceived quality [...] we must ask the people who are actually there.”

FG-STO-P4: “Quality involves many aspects, including the care model and how the organization conceptualizes care. It’s about whether the person is genuinely placed at the center and how everyone works around them. It also involves training professionals to effectively assess and improve residents’ quality of life. Quality is multifaceted and viewed from various perspectives.”

### 3.12. Emerging Issues

Based on the transcriptions from the FGs, three fundamental emerging issues arose that influence both the quality of residential centers and the QoL of older adults residing in them. These issues, labeled “E1. Adapted to the user profile”, “E2. Small residential center”, and “E3. Knowledge of the sector”, reflect concerns and observations shared by the participants in the FGs. Each of these emerging issues is defined and discussed below.

The analysis is based on the co-occurrence table of codes related to the emerging issues concerning the various stakeholders from the three FGs (Figure 4 and Figure 5).

From Figure 4 and Figure 5 it can be inferred that there was a greater emphasis on the issue of adapting to the user profile (E1) by the FG-MTS group, suggesting a higher sensitivity of this group to the specific needs of the residents. Additionally, all three focus groups mentioned the issue of small residential centers (E2) with similar frequency, reflecting a widespread consensus on the benefits of the small residential center model. In contrast, the importance of in-depth knowledge of the sector (E3) generated a considerable number of responses, especially from the FG-MTS group.

### 3.13. E1. Adapted to the User Profile

This first emerging issue refers to the necessity of adapting the services and spaces of residential centers to the individual characteristics of users. An example provided in the transcriptions illustrates how a group of caregivers in a European project decided to modify the physical environment of the center to meet the needs of residents with visual impairments. By utilizing strong and light color contrasts in everyday elements such as plates and hallways, they succeeded in enhancing the autonomy of the residents, who required less support for mobility and daily activities.

FG-MTS-P2: “[...] I will give you an example of something I would have loved for us to implement, but it isn’t; it’s from another group of residential companies. They presented a project at the European level, and their team of occupational therapists decided that there were many individuals in their center with visual impairments, which is quite common and often accompanies other issues. They conducted research to ensure that the spaces were consistently contrasted with strong and light colors, from plates to forks, making sure each item had a color and even the layout of the hallways. They found that individuals with visual impairments became more autonomous and required less assistance both for moving around and for engaging in activities such as eating.”

The key to this approach is recognizing that treatments and solutions cannot be generalized for all residents. Participants emphasized that care must be personalized and tailored to the specific conditions of each individual. As one participant mentioned,

FG-MTS-P3: “[...] it should be a bit more adapted to the profile of each of the users we ultimately work with in our center […]”

This reflection underscores that adapting the centers to individual needs not only enhances the quality of the center but also improves the QoL of residents, fostering their independence and autonomy.

### 3.14. E2. Small Residential Centers

The second emerging issue highlights the advantages and challenges associated with small residential centers. According to the transcriptions, smaller centers facilitate direct and personalized attention to residents. Participants noted that in these environments, staff members develop closer relationships with residents, which fosters a deeper understanding of their preferences and needs. As expressed in one of the testimonials,

FG-MTS-P4: “[...] in a small center, residents can be attended to much better because, ultimately, the facilities themselves are smaller, allowing you to reach and access them more quickly.”

However, despite these benefits, challenges were also mentioned. For instance, some management aspects, such as clothing choices and meal planning, must be organized centrally to avoid logistical issues and ensure resource availability. One participant remarked,

FG-MTS-P7: “[...] although it is a small center, it is very difficult to control... so many things cannot be managed.”

This comment reflects the operational limitations that can arise even in smaller environments, suggesting that a balance between flexibility and control is necessary to maximize the benefits of small centers.

### 3.15. E3. Knowledge of the Sector

Finally, the third emerging issue addresses the importance of sector knowledge among the professionals managing and working in residential centers. Participants emphasized the need for detailed study and greater attention to established regulations and procedures for caring for older adults. A disparity was observed in the interpretation and application of these regulations, leading to inconsistencies in care standards. As one participant expressed,

FG-MTS-P3: “[...] each person who comes has a different set of regulations... it seems unbelievable, because the regulations are the same, but everyone interprets them differently.”

FG-PP-P5: “[...] I believe that a person who has been stuck in an office since they turned 23 is not the one to regulate what a center should have and what it should not have.”

This fragmentation in understanding regulations can create confusion among both staff and residents, potentially impacting service quality. Professionals call for greater coherence and clarity in regulatory applications, as well as a more collaborative approach that includes direct contact with professionals in decision-making and the implementation of best practices.

## 4. Discussion

The development of this research has involved the collection of data through various methods, enhancing the scientific rigor and findings of previous studies. FGs provided crucial insights into the current characteristics of residential centers in Spain, as well as potential improvements within the system. Within these groups, key themes have emerged, including knowledge of the sector and a preference for small, user-profile-adapted centers. This aligns with studies demonstrating improved QoL in such environments [38]. Participants expressed a preference for smaller centers or living units in familial settings for residents, underscoring the need to preserve their social and personal connections.

The analysis reveals that the ownership of the center (public or private) does not impact the quality of care, consistent with previous studies [17,44,45,46]. Furthermore, the literature suggests that there are insufficient tools to effectively measure QoL in centers, considering both objective and subjective aspects [24]. Research highlights the importance of a comprehensive approach that includes residents, families, and professionals, as well as cultural and regional factors affecting QoL. In Spain, longevity and active lifestyles are factors that must also be considered when assessing the QoL of residents [46].

Based on the findings from the FGs, it is crucial to improve the spaces, services, and human resources of the centers to provide individualized and homely care. The preferences and needs of residents should be the central focus of attention. Despite geographic and sociodemographic differences, the research does not provide reliable data to compare how the characteristics of centers affect QoL. There is a recognized need for more residential placements and strategies that promote autonomy and social integration for older adults [47].

In Spain, the familistic model, where families assume the care of older adults, is outdated, necessitating greater state intervention in the provision of care services [48,49]. Education should focus on promoting active aging and a fulfilling life for older adults, addressing both independent individuals and individuals dependent on care.

Conceptual models of QoL must be adapted to the Spanish context to enhance resource utilization in residential centers [50]. Research in Spain on QoL in these centers is limited, underscoring the need for an advanced QoL model that encompasses the particularities of the resident population [25].

The emerging issues identified from the FGs highlight the complexity of managing residential centers that can offer both quality facilities and enhance the lives of residents. Service personalization (E1) is crucial for promoting autonomy and well-being among residents, while smaller centers (E2) allow for more direct interactions, although they must be managed carefully to avoid operational challenges. Lastly, coherent knowledge and application of regulations (E3) are essential to ensure that centers operate efficiently and effectively.

These issues emphasize the need for a multidimensional approach to improve both the quality of residential centers and the QoL of residents. It is evident that there is no one-size-fits-all solution; each center must tailor its strategies to the characteristics of its population and the specific conditions of its environment. However, by addressing these emerging issues, key areas for improvement can be identified, contributing to the enhancement of sector standards and the well-being of older individuals living in these centers.

This study involved certain considerations related to the use of focus groups. The political profile focus group, consisting of five participants, provided valuable insights despite the challenges of scheduling and the participants’ hesitation to share sensitive information in a group setting. Additionally, obtaining a geographically diverse sample from across the country proved difficult. To address this, Google Meet was utilized for virtual meetings, facilitating broader participation while overcoming some logistical barriers.

Initially, we considered conducting the study using questionnaires directed at the residents, family members, and staff of the centers. However, two limitations arose: first, the pandemic hindered direct access for information gathering; second was the variety of measurement scales in the scientific literature and the divergence in the aspects each one measures.

The QoL model for older adults residing in these centers should integrate health, psychological, and social variables from both residents and their families and staff. These models should be part of the QoL assessment for the older population. It is crucial to continue implementing new residential models that prioritize the opinions and desires of older adults, moving towards a personalized and inclusive approach, with specific dimensions for evaluating QoL based on each individual’s circumstances and experiences.

A limitation of this study, however, is that the evaluation of QoL is based on the perspective of professionals and does not include the direct voice of the residents. While method triangulation has been used to obtain a more comprehensive view, it is important to recognize that residents’ perceptions of their own QoL are essential. Future studies should incorporate the perspective of residents through methods that allow them to express their experiences and needs individually.

Finally, the cultural change movement in residential centers, initiated in the late 1980s to improve the QoL of older adults and foster connections with staff, has made limited progress over more than 40 years. The global pandemic highlighted the need for residential centers to be versatile spaces capable of adapting to various circumstances, where bonds are formed through daily interactions and support. However, there are temporal limitations and a lack of active policies to promote effective change in these centers. Furthermore, this cultural shift is complex and requires the involvement of all social sectors, making it difficult to achieve.

## 5. Conclusions

This study reveals a complex reality that is growing daily and demands effective intervention policies, not only in terms of increasing the number of residential placements and renewing facilities but also in improving the care provided to older adults. The term “accompaniment” has been key in this research, suggesting that supporting older adults, as well as technical staff and political sectors, can generate a chain of benefits for all parties involved. This would facilitate improved quality in services and centers, promoting person-centered care and establishing new quality standards. Such changes involve renewing infrastructures and creating multifunctional spaces that promote well-being and inclusion, opening centers to the community. Participants in the FGs emphasized the need for centers to be open and focused on human relationships, both with residents and with staff and the community.

The existing literature and research indicate that the introduction of new models improves the well-being and QoL of older adults in residential settings; however, there is a lack of studies addressing all aspects of residential centers and their relationship with QoL. More research is needed to corroborate these findings and to study the relationship between QoL and management characteristics, facilities, and services. Additionally, further investigation is required on structural aspects, financial resources, and personnel management to better understand their relationship with QoL.

Furthermore, studies should expand to more geographic areas to enable managers and policymakers to enhance healthcare policies, influencing the design and characteristics of centers from their initial stages. Cultural change in care centers, centered on older adults, is crucial for improving the daily lives of residents and their integration into society. Additional research is needed that reflects the desires and needs of older adults and their families concerning their environment, facilitating understanding of how centers operate and promoting tools to prevent illness and enhance well-being. Moreover, future studies should emphasize the inclusion of both residents and their families, ensuring that their voices are heard and their perspectives are considered. Comparing the insights provided by staff with those of residents and their families will provide a more comprehensive understanding of the dynamics within care centers, ultimately supporting more effective strategies for improvement.

The FG study concludes that cultural change in the residential model does not depend on public or private management, or the size or location of centers, but rather on ensuring quality care, proper training of staff, and adequate funding of the system. Moreover, a structural renewal of centers across the country is necessary to equitably meet the needs of the population in all regions, in response to demographic changes and the need to focus on the preferences of residents.

## Figures and Tables

**Figure 1 healthcare-12-02446-f001:**
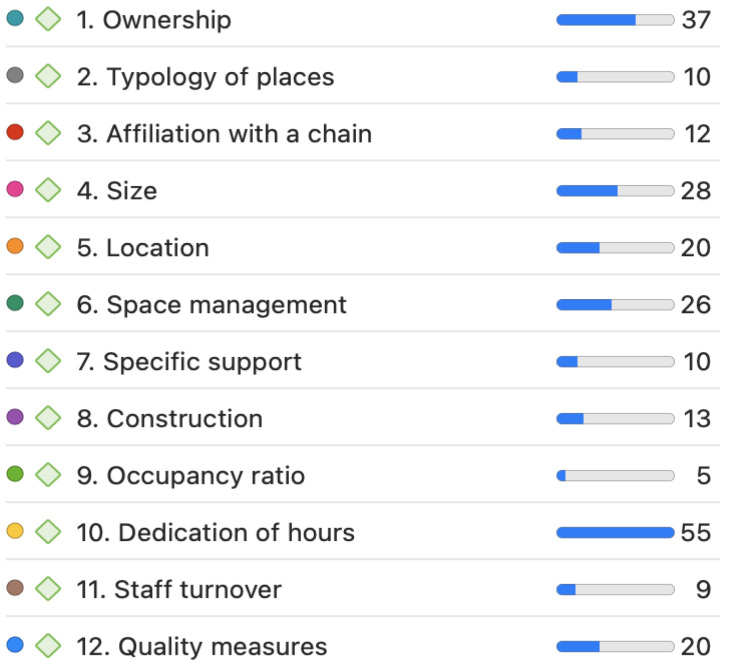
Codes associated with questions from FGs (Appendix A) and frequency of quotations addressing these questions, with the color of each code corresponding to those used in the Sankey diagrams. Source: Own elaboration.

**Figure 2 healthcare-12-02446-f002:**
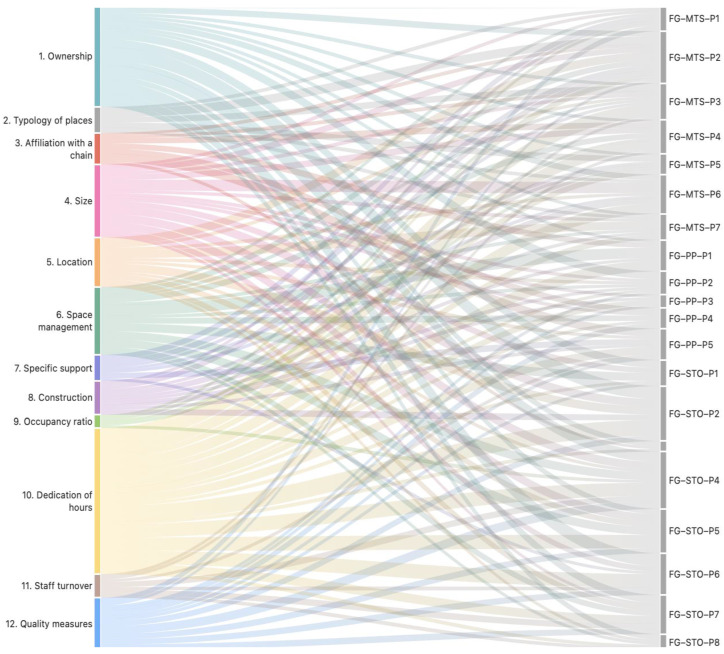
Sankey diagram of co-occurrence of codes for questions and intervening individuals from the FGs. Source: Own elaboration.

**Figure 3 healthcare-12-02446-f003:**
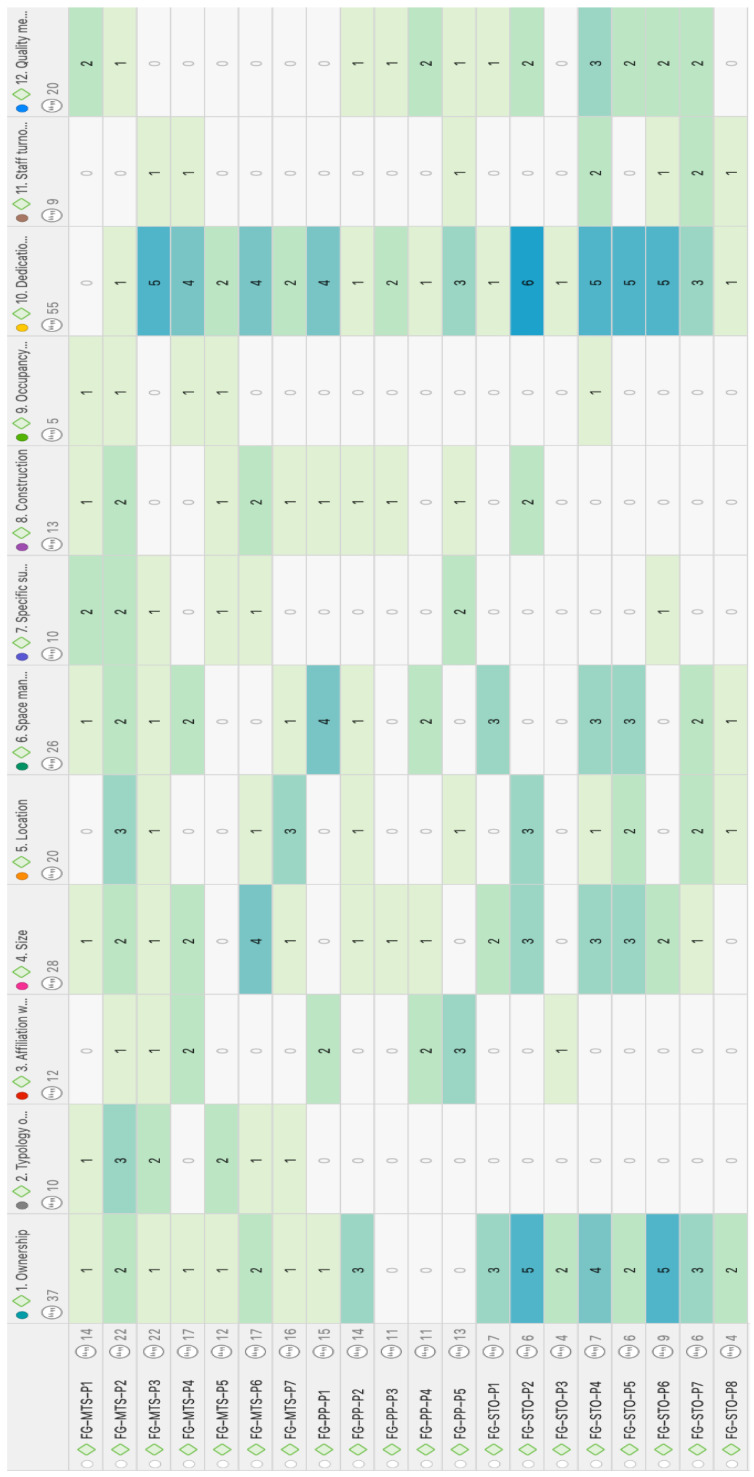
Co-occurrence of codes for questions and intervening individuals from the FGs. Each row represents a participant from the focus groups. Each column corresponds to a question code. The intensity of the color in each cell indicates the frequency of responses or co-occurrence of codes, with darker colors representing higher frequency. Source: Own elaboration.

**Figure 4 healthcare-12-02446-f004:**
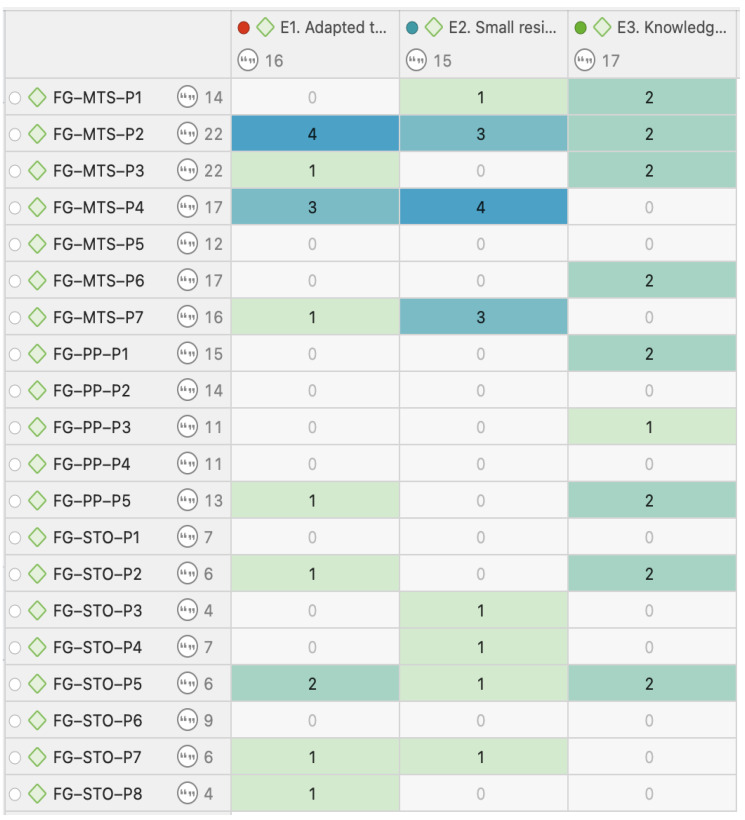
Co-occurrence of codes for emerging issues and intervening individuals from the FGs. Each row represents a participant from the focus groups. Each column corresponds to a question code for emerging issues. The intensity of the color in each cell indicates the frequency of responses or co-occurrence of codes, with darker colors representing higher frequency. Source: Own elaboration.

**Figure 5 healthcare-12-02446-f005:**
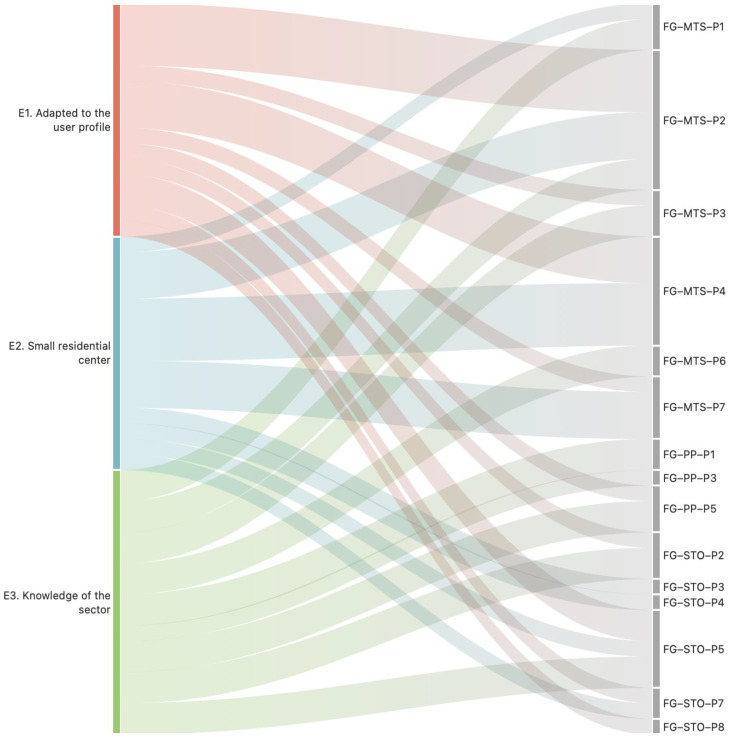
Sankey diagram of co-occurrence of codes for emerging issues and intervening individuals from the FGs. Source: Own elaboration.

**Table 1 healthcare-12-02446-t001:** Sociodemographic data of participants.

Code	Gender	Age	Position/Profession
FG-MTS-P1	Female	46	Social Worker
FG-MTS-P2	Female	44	Social Worker
FG-MTS-P3	Male	51	Head of Nursing
FG-MTS-P4	Female	34	Social Worker
FG-MTS-P5	Male	48	Director
FG-MTS-P6	Female	53	Director/Clinical Assistant
FG-MTS-P7	Female	37	Director/Clinical Assistant
FG-STO-P1	Male	48	Service Chief
FG-STO-P2	Male	55	Managing Director
FG-STO-P3	Female	42	Social Worker
FG-STO-P4	Female	68	General Director
FG-STO-P5	Male	54	Medical Director
FG-STO-P6	Male	66	Banker
FG-STO-P7	Male	48	Director
FG-STO-P8	Male	56	Manager
FG-PP-P1	Male	38	Operations Director
FG-PP-P2	Male	52	Lawyer
FG-PP-P3	Female	58	Psychologist
FG-PP-P4	Female	51	Law Graduate
FG-PP-P5	Male	54	Social Educator

Source: Own elaboration.

## Data Availability

The qualitative dataset and transcriptions of narratives are not publicly available due to ethical restrictions and privacy issues.

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
