# Peer review of "Life and Quality of Life of Older Adults in Residential Centers in Spain: A Focus Group Analysis Based on Center Quality from the Perspective of Management Stakeholders"

_healthcare, 2024, doi:10.3390/healthcare12232446_

Round 1
Reviewer 1 Report
Comments and Suggestions for Authors
great work with this manuscript and study
I added many comments inside the text
in particular about the following aspects: rigour of the thematic analysis, description of themes,, meaning of tables and figures and minimizing the repetitive material.
I guess you have it in mind but it is not clearly presented.

Reviewer 2 Report
Comments and Suggestions for Authors
This is a very topical area to consider and is of international concern. You have done a lot of work, but I feel the value of your work is diminished due to the writing style and structure of the article.
The article is too wordy and there is a lot of unnecessary repetition, some of which I have suggested below. You need to consider the structure of your article to make it easy for the reader to follow a logical flow.
The introduction needs to present what is known on this topic and provide a logical argument to support the aims of your research.
Methods need to be detailed enough to tell the audience what you did, but as I mention you have to assume that your readership will have some professional knowledge. The methods are too long! You can briefly say why you chose certain software to use for analysis, but we do not need a paragraph on the features of the software.
The results section should not contain new material. It needs to present in logical order what you found. You also cannot present the results of an already published literature review here. While it may underpin methods in your study, this needs to go into the introduction. Tables and figures need to enhance the readers understanding of the material. You tables need review to label axes in a meaningful way, and a key to understand what the various colours may mean. I am sorry , but I just found them confusing rather than illuminating!
Discussion Genrally well presented, however as QoL was a major part of your aims, I feel this should have been explored in the introduction, not giving definitional concerns as a limitation. Having said this another limitation is the question around QoL, as seen by a third party is questionable.
Conclusions. Your conclusions are brief and well presented. I would like to feel confident that the results back up your conclusions. The results need a major rewrite to enable reader to appreciate that work you have done.
Some specific comments.
Abstract and Title
Well written and clear, though QoL of older adults is usually investigated by direct methods with the older people themselves, which is what I expected from the title. It appears from just reading the abstract that you are investigating QoL from the perspectives of professional staff.
Please consider making this approach explicit in the title.
Introduction
P1, line 37 the phrase “expected to multiply by 2067” can you be more specific with this projection?
Residential care centre could be more defined. Are these nursing homes or the equivalent of an American long term care centre or UK residential care home? Maybe some idea of who is admitted (eg function, cognitive or physical) will help orient the reader.
p.2 line 56 the word order in English needs a small change for clarity , from “….establishes the right to care for dependent individuals’ to “ the right of older individuals to receive care” and maybe further clarify they level or type of care such as “for their assessed needs” or whatever is appropriate and accurate.
The aims are wordy – try to make them brief and to the point. If some are aims and some objectives, be explicit and let the formatting of the text help make this clear.
Materials and methods
This is really just methods as you don’t describe any materials as such. ( I am not including interview design or analysis software as materials).
This whole section could be shortened. For example, while interesting the history of focus groups is not central to your paper and I would leave this out eg. lines 99-101 as one example. Most readers of this journal would be familiar with this qualitative technique.
P.2 Line 75 say what sort of analysis you did eg content, thematic, grounded, narrative analysis etc
2.1 Participants. please consider making this section briefer. You mention local, regional and national many times!
Write out Focus Group – Management and Technical Staff in full first, then provide the abbreviation FG-MTS. Same for the other two focus groups
2.2 the first sentence ( line 151) is redundant.
As I continue reading I am finding the text is too wordy!
As a journal article for a professional audience you have to assume some level of knowledge. In scientific writing, brevity is best!
For example
For the qualitative analysis, ATLAS.ti version 23 software was employed, chosen for its user-friendly interface and capability to handle various formats of primary data. It is one of the most widely used software tools for theory construction. This software is particularly effective for establishing connections among different data elements, allowing for the association of codes or labels with text segments, identifying patterns, and constructing classifications of codes that reflect testable models of the underlying conceptual structure of the data [33,34].
Could simply be
ATLAS.ti (V23) was used for data analysis and structuring of codes to identify patterns within the data. (Or something similar)
2.3 procedure.
Information about recruitment needs to be in the participant section
2.4 this section is redundant – you have already mentioned this in sections above.
Results
You cannot present results of your previously published literature review here, nor in the methods. Please include this as part of the introduction. You can refer to the literature in your discussion as well, but please restrict your results to the current study only. this will also mean you need to amend your Methods section.
Please reconsider your use of tables. The labelling of axes eg FG-MTS-P1 is not meaningful nor do you provide a key for the colour coding in the table.
Further information in the introduction about the care facilities would be helpful for the reader to understand results. For example the sentence p. 10 line 324 – 326 is the first time you introduce the idea that there are publicly funded and private pay residents in the same facility.
Subheadings would be helpful in this section. I am getting the impression that the qual results are presented as per the coding structure presented in Fig 1.
You can consider a re structure of the narrative results under sub headings and leave out the figure.
Keep illustrative quotes to one or two pertinent ones. Highlight where there are discrepancies between different groups. Ensure that quotes support statements in the narrative. For example in the section on construction and location it sounds like this is what you are saying as authors, not what arose from the focus groups – you mention centres being on the outskirts of the city, but the quotes don’t mention this.
